# Facile Assembly of InVO_4_/TiO_2_ Heterojunction for Enhanced Photo-Oxidation of Benzyl Alcohol

**DOI:** 10.3390/nano12091544

**Published:** 2022-05-03

**Authors:** Xinyu Zhang, Quanquan Shi, Xin Liu, Jingmei Li, Hui Xu, Hongjing Ding, Gao Li

**Affiliations:** 1College of Science, Inner Mongolia Agricultural University, Hohhot 010018, China; zhangxy04@yeah.net (X.Z.); qqshi@dicp.ac.cn (Q.S.); ljm19970043@163.com (J.L.); helloxiaoding@126.com (H.D.); 2State Key Laboratory of Catalysis, Dalian Institute of Chemical Physics, Chinese Academy of Sciences, Dalian 116023, China; 3Institute of Advanced Materials, College of Chemistry and Chemical Engineering, Jiangxi Normal University, Nanchang 330022, China

**Keywords:** InVO_4_/TiO_2_ heterojunction, photocatalysis, oxidation, benzyl alcohol

## Abstract

In this work, an InVO_4_/TiO_2_ heterojunction composite catalyst was successfully synthesized through a facile hydrothermal method. The structural and optical characteristics of InVO_4_/TiO_2_ heterojunction composites are investigated using a variety of techniques, including powder X-ray diffraction (XRD), transmission electron microscopy (TEM), and spectroscopy techniques. The addition of InVO_4_ to TiO_2_ considerably enhanced the photocatalytic performance in selective photo-oxidation of benzyl alcohol (BA). The 10 wt% InVO_4/_TiO_2_ composite photocatalyst provided a decent 100% BA conversion with over 99% selectivity for benzaldehyde, and exhibited a maximum conversion rate of 3.03 mmol g^−1^ h^−1^, which is substantially higher than bare InVO_4_ and TiO_2_. The excellent catalytic activity of the InVO_4/_TiO_2_ photocatalyst is associated with the successful assembly of heterostructures, which promotes the charge separation and transfer between InVO_4_ and TiO_2_.

## 1. Introduction

The consumption of chemical fuels results in ever-increasing environmental and energy issues; thus, developing a green and renewable energy source is essential [1,2,3]. Selective oxidation has been widely recognized as a promising strategy for alcohols to produce corresponding aldehydes, especially from benzyl alcohol toward benzaldehyde, which is one of the most important chemical processes in industry [4,5]. Benzaldehyde (BAD), one of the most common industrial chemicals, is widely used in the manufacturing of pharmaceuticals, perfumes, dyes, and additives [6,7]. However, several traditional thermal synthesized routes, such as the oxidation of toluene and hydrolysis of benzyl chloride, use high-valent metal salts (e.g., permanganate, dichromate) and lead to a variety of hazardous wastes [8]. Photocatalytic selective oxidation benzyl alcohol (BA) using O_2_ as a green oxidant is recognized as a promising strategy for replacing traditional materials [9,10].

Among various semiconductor photocatalysts, TiO_2_ has been extensively studied owing to lower cost, nontoxicity, and exceptional photocatalytic performance [11]. Nevertheless, the wider bandgap (only absorbs ultraviolet) and higher recombination rate of photogenerated electrons and holes seriously restrict its photocatalytic activity under sunlight [12]. The heterostructure composite is regarded as one of the most promising ways for harvesting sunlight and boosting the photocatalytic activity with the synergistic effects of heterogeneous interfaces between multiple semiconductors [13,14,15]. Oliveira et al. prepared heterojunction clusters of M@Si_12_ (M = Ti, Cr, Zr, Mo, Ru, PD, HF and OS) and M@Si_16_ (M = Ti, Zr, Hf) and studied their properties [13].

Recently, InVO_4_ with a bandgap of 2.0 eV has been indicated as a novel visible-light-responsive photocatalyst [16]. The orthorhombic crystalline phase of InVO_4_ consists of VO_4_ tetrahedra sharing corners with InO_6_ groups to form compact In_4_O_6_ groups [17], which recently was used for various photocatalytic reactions, such as H_2_ evolution and photodegradation of organic pollutants [16,18]. However, the activity of the InVO_4_ photocatalyst is not ideal due to sluggish electron mobility. Motivated by these points, we herein successfully constructed InVO_4_/TiO_2_ heterojunction composites via an in situ hydrothermal transformation of titanate nanotubes with indium nitrate, which is verified through powder X-ray diffraction, transmission electron microscopy, and spectroscopy techniques. The InVO_4_(112)/TiO_2_(101) heterogeneous interface largely promotes the separation and transfer of photogenerated charges between InVO_4_ and TiO_2_. A 10 wt% InVO_4_/TiO_2_ photocatalyst showed a maximum conversion rate of 3.03 mmol g^−1^ h^−1^. Furthermore, trapping experiments verified that the active radicals of ·O_2_^−^ ·OH and h^+^ are responsible for the enhanced activity. Overall, this work may provide important information for the exploration of new materials for photocatalytic oxidation of BA to BAD.

## 2. Materials and Methods

### 2.1. Preparation of Sodium Titanate Nanotube Precursor

All chemicals were commercially available as reagent grade and were used as received without further purification. Titanate nanotubes were synthesized via a facile hydrothermal method. Initially, 12 g NaOH was dissolved into 30 mL distilled water to form 10 mol L^−1^ NaOH aqueous solution in a 50 mL Teflon bottle, and 1 g P25 was added. After stirring for 30 min, the Teflon bottle was sealed tightly and heated at 120 °C for 24 h. The precipitate was centrifuged and washed with distilled water several times until the pH reached about 8. The obtained white precipitates were wet sodium titanate nanotubes.

### 2.2. Preparation of InVO_4_/TiO_2_ Heterojunction Compound Catalyst

InVO_4_/TiO_2_ composite photocatalysts were prepared via a simple hydrothermal reaction of sodium titanate nanotubes with InCl_3_ in the presence of NH_4_VO_3_. Typically, a certain proportion of NH_4_VO_3_ and InCl_3_ was dissolved in 30 mL distilled water, and then pH was adjusted to 1–2 using nitric acid. Next, 1 g of sodium titanate nanotubes was dispersed to the former suspension under vigorous stirring. After 30 min, the slurry was transferred into autoclaves and heated at 180 °C for 18 h. The obtained precipitate was washed with water and dried at 80 °C for 12 h. Additionally, by controlling additional amounts of InCl_3_ and the titanate nanotubes, different ratios of InVO_4_/TiO_2_ heterojunction composite catalysts were obtained and denoted as InTi-x (x represents the weight amount of InVO_4_, e.g., 5, 10, and 15). The InVO_4_ samples were prepared via similar procedures in the absence of titanate nanotubes.

### 2.3. Catalyst Characterization

Powder X-ray diffraction (XRD) patterns were performed using a PANalytical Empyream diffractometer operated at 40 kV and 200 mA. Transmission electron microscopy (TEM) images were recorded using field emission transmission electron microscopy (FEI Tecnai F20). Specific surface areas were determined using nitrogen adsorption at 77K and the BET method. Pore size distributions were derived from nitrogen desorption isotherms using the Barrett-Joyner-Halenda (BJH) method. Fourier transform-infrared (FTIR) spectra were obtained using a Perkin-Elmer Spectrum 100 Spectrometer at room temperature. X-ray photoelectron spectra (XPS) were tested on an ESCALAB MK-II spectrometer (VG Scientific Ltd., London, UK) with Al Kα radiation. UV-visible diffuse reflectance spectra were recorded in the range of 200–800 nm on a spectrophotometer (PE Lambda 850) using BaSO_4_ as a reference. Photoluminescence (PL) spectra were operated at room temperature on a fluorescence lifetime spectrophotometer (Hitach, F4600, Japan). The photoelectrochemical tests of samples were carried out on an electrochemical workstation (CHI760E, Shanghai) with a standard three-electrode system, including an Ag/AgCl electrode and a Pt plate as the reference and counter electrodes. The as-obtained samples were coated on FTO glass substrates to be used as the working electrodes. A 300 W Xe lamp with a cut-off filter (λ > 420 nm) was used as the light source, and all of the electrochemical tests were carried out in KH_2_PO_3_-K_2_HPO_3_ buffer solution (0.5 M, pH = 7).

### 2.4. Catalytic Tests

Photocatalytic oxidation of benzyl alcohol was carried out in a stainless steel autoclave. Typically, 50 mg of catalyst, 40 μL of benzyl alcohol, and 20 mL of acetonitrile were added to a reactor. Next, the autoclave was filled with pure oxygen of 0.5 MPa and then was irradiated with a 300 W xenon lamp at a stirring rate of 500 rpm. After the reaction, the mixture was centrifuged to obtain the supernatant, which was analyzed by gas chromatographic (Agilent 7820 GC) analysis. Furthermore in the cycle stability test, the used catalysts were centrifuged, washed, and dried for the next run. In a free radical capture experiment, 20 μL Isopropanol (IPA), ammonium oxalate (AO), and 1,4-benzoquinone (BQ) were added respectively as scavengers of hydroxyl radical (·OH), hole (H^+^) and superoxide radical (·O_2_^−^).

## 3. Results

### 3.1. Synthesis and Characterization of InVO_4_/TiO_2_ Nanocomposites

InVO_4_/TiO_2_ nanocomposites were prepared based on the Na_2_Ti_3_O_7_, nanotubes were dissociated and transformed into TiO_2_ nanoparticles, and the InVO_4_ nanoparticles were simultaneously grown and anchored onto the TiO_2_ surface. The phase composition and crystal structure of the prepared sample were characterized by power XRD. Figure 1a shows that the diffraction peaks of pure TiO_2_ could be indexed to anatase-TiO_2_ [19]. Bare InVO_4_ could be indexed to the orthonormal phase with the diffraction peaks at 18.5, 20.8, 31.1, 33.0, 41.6, 51.0, and 60.9°, corresponding to (110), (020), (200), (112), (202), (042), and (242) crystal facets of InVO_4_ (JCPDS 48-0898) [20]. For the InTi-x composites, all diffraction lines match well with the feature lines of InVO_4_ and TiO_2_, and no diffraction peaks of other species were found, which indicates the formation of a heterojunction structure.

Figure 1b shows the Infrared spectra of TiO_2_, InVO_4_, and InTi-x. IR peaks of TiO_2_ at 3260 cm^−1^ and 1655 cm^−1^ were ascribed to O-H (tensile mode) and Ti-OH (bending mode), respectively [21]. Pure InVO_4_ displayed significant peaks at 896 cm^−1^ and 631 cm^−1^, which were related to V-O and V-O-In bonds, respectively [22,23]. The IR peaks of InTi-x composites were consistent with vibration peak of InVO_4_ and TiO_2_, proving the successful coupling of TiO_2_ and InVO_4_. Further, the InTi-10 sample exhibited a typical IV isotherm and a large surface area of 285.4 m^2^ g^−^^1^ (Figure 1c) [24].

The morphological characteristics of InVO_4_/TiO_2_ composite were studied by TEM (Figure 2). Figure 2a showed that InTi-10 composite is composed of particles with diameters of 10–20 nm. In addition, TiO_2_ and InVO_4_ were evenly distributed, and abnormally large particles were not observed. Figure 2b clearly shows the interface between TiO_2_ and InVO_4_, where the lattice spacing of TiO_2_ and InVO_4_ was 0.35 nm and 0.27 nm, corresponding to TiO_2_(101) and InVO_4_(112) planes, respectively [25]. The morphology of connected particles between InVO_4_ and TiO_2_ indicated the existence of a InVO_4_/TiO_2_ heterojunction.

X-ray photoelectron spectroscopy (XPS) was employed to investigate the valence states of TiO_2_, InVO_4_, and InTi-10 composites. Figure 3 depicts the XP spectra of In3d, V2p, Ti2p, and O1s. Figure 3a shows two distinct peaks at 452.8 eV (In 3d_5/2_) and 445.2 eV (In 3d_3/2_), associated with In^3+^ [26]. Two characteristic binding energies (BEs) at 524.9 eV and 517.4 eV in InVO_4_ correspond to V 2p_3/2_ and V 2p_1/2_, respectively, which are related to V^4+^ species (Figure 3a) [27,28]. For InTi-10 composites, slight shifts were seen, which indicate a strong interaction between InVO_4_ and TiO_2_. Two binding energies (BEs) were found at 458.5 eV and 464.2 eV in both TiO_2_ samples, which are assigned to Ti 2p_3/2_ and Ti 2p_1/2_ and correspond to Ti^4+^ species (Figure 3c) [29,30]. It was also noted that two characteristic peaks for InTi-10 composites could be identified; the peaks at 457.4 eV (Ti 2p_3/2_) and 463.1 eV (Ti 2p_1/2_) are assigned to Ti^3+^ [5]. The O 1s spectrum in the TiO_2_ sample consists of two types of oxygen species. The BEs at 529.9 and 531.4 eV (Figure 3d) are ascribed to lattice oxygen (O_L_) and oxygen vacancies (Ov), respectively [31,32,33]. Moreover, another oxygen species for bare InVO_4_ and InTi-10 composites have appeared with BEs at 532.1 eV, which correspond to hydroxyl groups (O_H_) on the oxide surface. Consequently, the obtained results revealed that InTi-10 composites exhibited a higher content of Ti^3+^ and Ov.

### 3.2. Optical Properties

The optical adsorption properties of TiO_2_, InVO_4_, and InTi-x composites were evaluated by UV-vis diffuse reflectance spectroscopy, as shown in Figure 4a. For bare TiO_2_, a strong adsorption band below 400 nm was observed, which showed the intrinsic larger band gap of anatase [34] compared with that of the bare ones. The absorption edge of InTi-x composites distinctively shifted to the visible region, and the intensity of light harvesting was also enhanced. Moreover, the band gap of TiO_2_, InVO_4_, and InTi-10 can be calculated through the converted Tauc plots, which are 3.48 eV, 2.18 eV, and 2.26 eV, respectively (Figure 4b). In addition, the Mott-Schotky measurements were investigated on TiO_2_ and InVO_4_ (Figure 4c,d), which indicated that both belong to the n-type semiconductor with the flatband potentials −0.25 and −0.37 V, respectively [35,36]. Finally, the conduction band edge can be calculated as 3.23 and 1.81 V, indicating a reasonable transfer of photogenerated charges occurred between TiO_2_ and InVO_4_ based the different energy band structure.

Photoluminescence (PL) spectroscopy was performed to reveal the separation efficiency of photogenerated charges, as shown in Figure 5a. The shape and position of emission peaks of InTi-x composites are similar to bare TiO_2_ and InVO_4_. Moreover, InTi-10 exhibited weakest the peak intensity, illustrating that the recombination of photogenerated electron-hole pairs is suppressed efficiently [37]. Figure 5b gives the photocurrent-time curves for TiO_2_, InVO_4_, and InTi-x composites. InTi-x composites showed enhanced photocurrent density compared with that of the bare TiO_2_ and InVO_4_ for several on–off cycles; moreover, InTi-10 exhibited the highest photocurrent density, indicating that InTi-10 can quite effectively promote the separation and transfer of photoinduced charge carriers [38]. Electrochemical impedance spectroscopy (EIS) was further conducted to study the transfer and separation efficiency of photogenerated charges, and the obtained Nyquist diagram and the fitting results are indicated in Figure 5c,d. Generally, each arc represents a resistance during the charge transfer process and a smaller radius associated with a lower charge-transfer resistance. InTi-10 composites had the smallest diameter and exhibited the fastest charge separation and transfer [39]. Therefore, it is reasonable to conclude that InTi-10 composites showed excellent photocatalytic performance based on these photo-response measurements.

### 3.3. Photo-Oxidation of Benzyl Alcohol

The catalytic performance of the as-prepared photocatalysts was conducted in photocatalytic oxidation benzyl alcohol (BA). The results are depicted in Figure 6a. Only the product of BAD was detected in the whole reaction process. It is worth mentioning that no BA conversion was found without light irradiation or catalysts. The bare TiO_2_ is almost inert in BA photocatalysis with a reaction of 3 h. After coupling with InVO_4_, all InTi-x composites exhibited higher performance. Among these catalysts, InTi-10 composites showed the best BA conversion (100%) and the highest formation rate (3.03 mmol g^−1^ h^−1^). The above result indicate that the content of InVO_4_ has an important effect on the photocatalytic performance, and the optimum percentage content is 10%. Furthermore, the cycle stability of InTi-10 composites for the photocatalytic oxidation of BA was also investigated, and no apparent loss appeared after four consecutive tests, indicating stable activity of the composite photo-catalysts. A radical trapping experiment was carried out on InTi-10 composites for the study of the main active species. Isopropanol (IPA), ammonium oxalate (AO), and benzoquinone (BQ) were used as scavengers for hydroxyl radicals (·OH), holes (h^+^), and superoxide radicals (·O_2−_), respectively [39]. As displayed in Figure 6c, the conversion of benzyl alcohol was significantly decreased after the addition of BQ, AO, and BQ, respectively implying that ·O_2_^−^, ·OH, and h^+^ should be the main catalytically active species for benzyl alcohol conversion.

Finally, a tentative reaction mechanism for aerobic oxidation of benzyl alcohol over InTi-x nanocomposites is proposed (Figure 7). The n-n heterojunction was formed at the interface between InVO_4_(200) and TiO_2_(110). Under built-in electric field driving, the photogenerated holes accumulated on the CB of TiO_2_ were transferred to CB of InVO_4_ for taking part in the oxidation of BA, while, the photoexcited electron easily transferred from VB of InVO_4_ to VB of TiO_2_ for reducing the adsorbed O_2_ of reactive oxygen species. Next, the adsorbed benzyl alcohol molecules on the surface of composites were oxidized by the photogenerated holes and ·O_2_^−^ species to result in the final product of benzaldehyde [40].

## 4. Discussion

In summary, we rationally designed and constructed a InVO_4_/TiO_2_ heterojunction composite via an in situ hydrothermal method. A series of characterization techniques were performed to indicate the successful fabrication of an n-n heterojunction. The InVO_4_(112)/TiO_2_(101) heterogeneous interface largely promoted the separation and transfer of photogenerated charges. Compared to bare TiO_2_ and InVO_4_, InTi-x composites exhibited a higher separation and transfer efficiency of the photogenerated charge and achieved a higher catalysis performance. InTi-10 composites gave a 100% benzyl alcohol conversion, with over 99% selectivity of benzaldehyde after a reaction of 3 h. This work is provides an efficient method for the design of photocatalytic composites for the photo-oxidation of alcohols to aldehydes.

## Figures and Tables

**Figure 1 nanomaterials-12-01544-f001:**
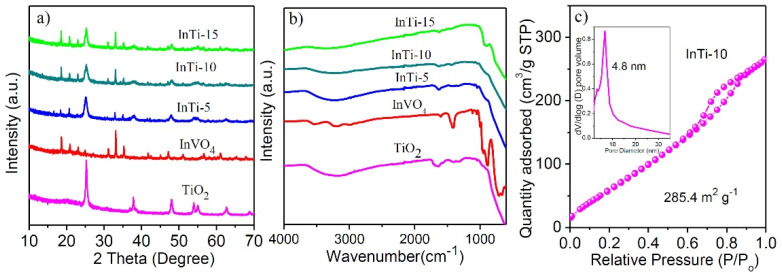
(**a**) XRD patterns and (**b**) FT-IR spectra of bare TiO_2_ and InVO_4_ and InTi-x composites. (**c**) N_2_-adsorption-desorption isotherms of InTi-10 composites.

**Figure 2 nanomaterials-12-01544-f002:**
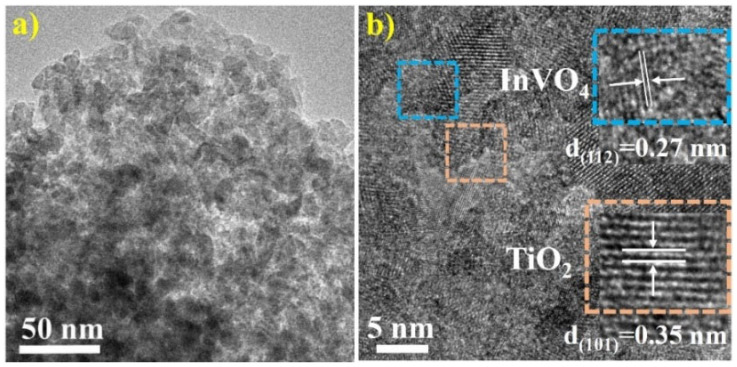
(**a**) TEM images and (**b**) HRTEM of InTi-10 composites.

**Figure 3 nanomaterials-12-01544-f003:**
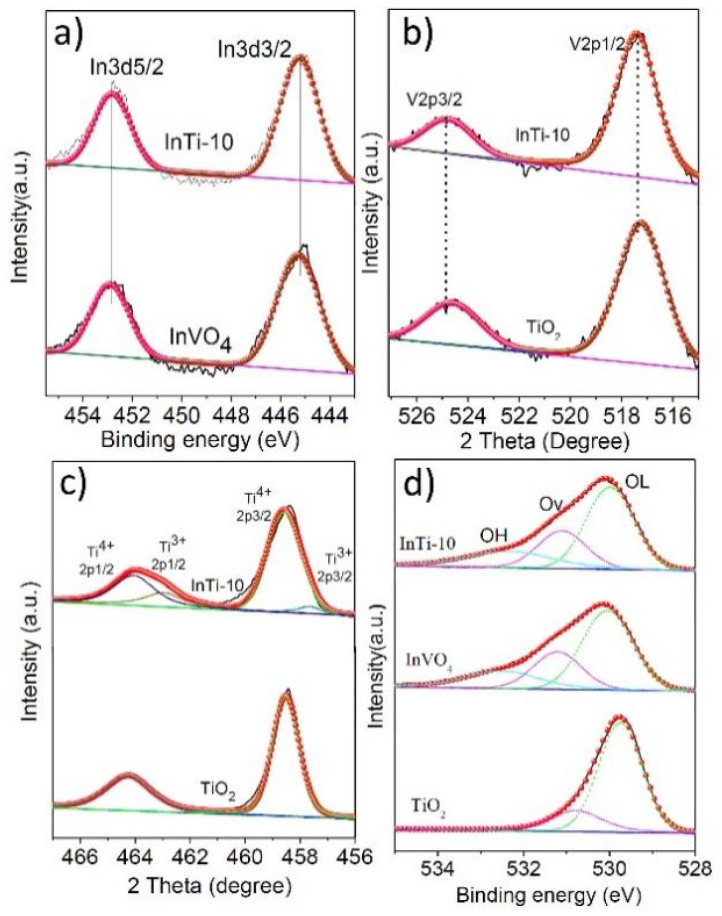
XPS peaks of InVO_4_, TiO_2_, and InTi-10 composites: (**a**) In 3d; (**b**) V 2p; (**c**) Ti 2p; (**d**) O1s.

**Figure 4 nanomaterials-12-01544-f004:**
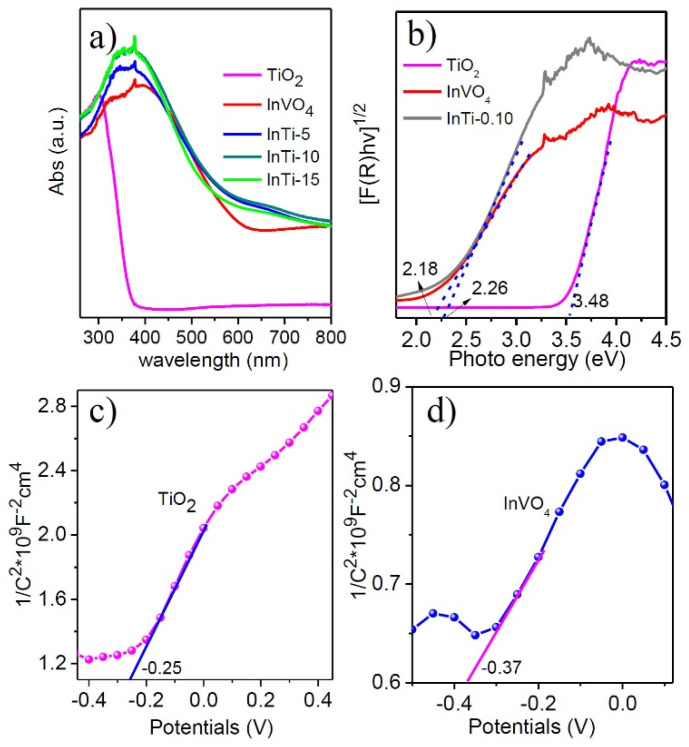
(**a**) UV-vis diffuse reflection spectra, (**b**) corresponding plots of (αhν)^1/2^ vs. photo energy of InVO_4_, TiO_2_, and InTi-x composites. Mott-Schottky plots of (**c**) TiO_2_ and (**d**) InVO_4_.

**Figure 5 nanomaterials-12-01544-f005:**
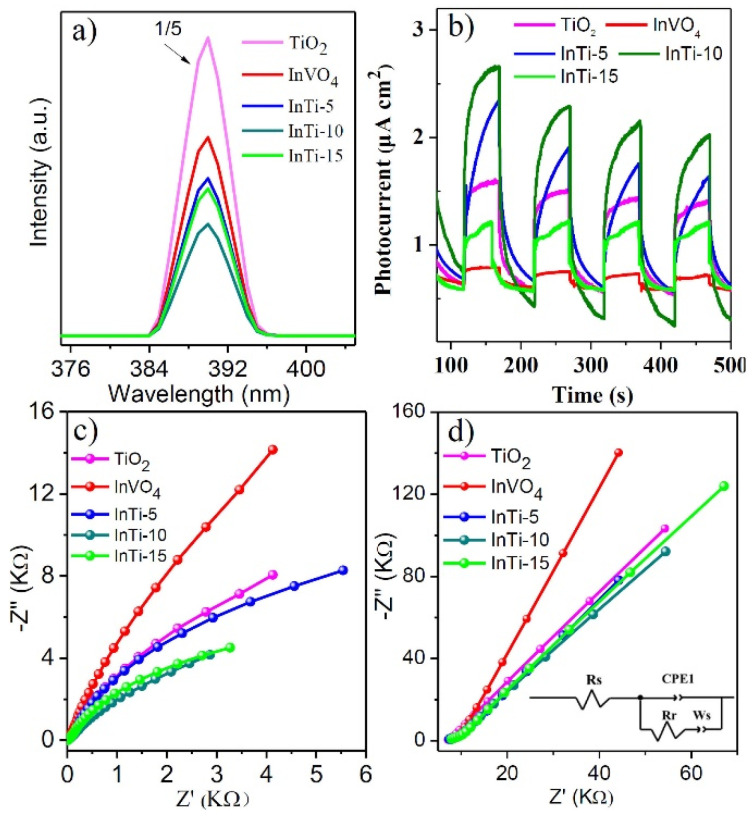
(**a**) PL spectra, (**b**) transient photocurrent spectra, (**c**) EIS Nyquist plots, and (**d**) impedance plots of InVO_4_, TiO_2_, and InVO_4_/TiO_2_ composites.

**Figure 6 nanomaterials-12-01544-f006:**
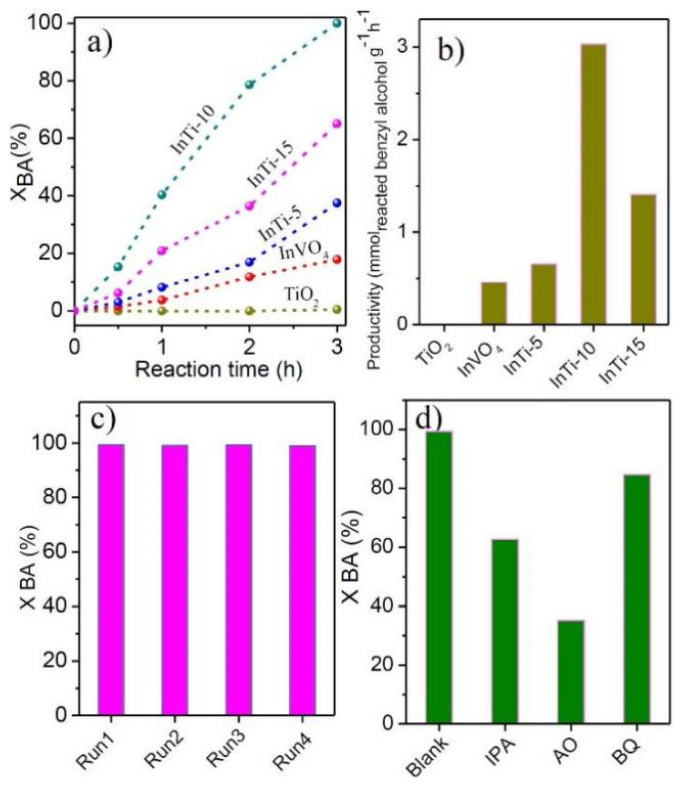
(**a**) Conversion of BA as a function of reaction time, and (**b**) productivity over InVO_4_, TiO_2_, and InVO_4_/TiO_2_ composites; (**c**) recyclability and (**d**) reactive species trapping tests over the InTi-10 photocatalyst.

**Figure 7 nanomaterials-12-01544-f007:**
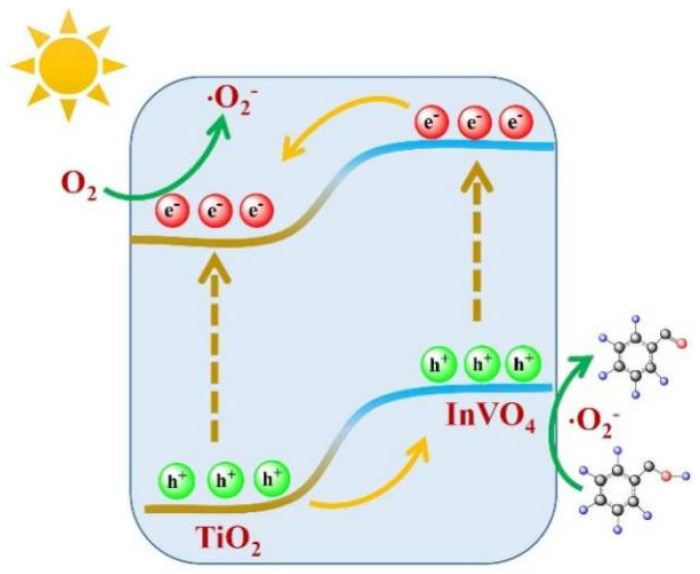
Schematic illustration of photoelectron–hole pair separation mechanism for InVO_4_/TiO_2_ under light illumination.

## Data Availability

Data are contained within the article.

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
