# Peer review of "Facile Assembly of InVO4/TiO2 Heterojunction for Enhanced Photo-Oxidation of Benzyl Alcohol"

_nanomaterials, 2022, doi:10.3390/nano12091544_

Round 1

Reviewer 1 Report

The paper is focused on  Facile assembly of InVO4/TiO2 heterojunction for enhanced photo-oxidation of benzyl alcohol.

In this work, the morphological characteristics of InVO4/TiO2 composite were studied by TEM. However, the authors must also investigate the powder morphology from BET analysis, such as surface area, total volume of pores and the mean pore diameter.

In electronchemical impedance spectroscopy (EIS) analysis the authors showed the Nyquist diagram and reported that: “each arc represents a resistance during the charge transfer process and a smaller radius are associated with a lower charge-transfer resistance.” The authors should further analyze of impedance spectroscopy data. The equivalent circuit must be determined from the fitting of the diagrams. The fitting provides the equivalent circuit and is possible to obtain the resistance and capacitance values of the charge transfer process.

Author Response

In this work, the morphological characteristics of InVO4/TiO2 composite were studied by TEM. However, the authors must also investigate the powder morphology from BET analysis, such as surface area, total volume of pores and the mean pore diameter.

Response: Thanks for Reviewer’s suggestion. We have added the BET test over InTi-10 composites, as shown in Figure 1c in the revised manuscript.

In electronchemical impedance spectroscopy (EIS) analysis the authors showed the Nyquist diagram and reported that: “each arc represents a resistance during the charge transfer process and a smaller radius are associated with a lower charge-transfer resistance.” The authors should further analyze of impedance spectroscopy data. The equivalent circuit must be determined from the fitting of the diagrams. The fitting provides the equivalent circuit and is possible to obtain the resistance and capacitance values of the charge transfer process.

Response: Thanks for Reviewer’s suggestion. We have added the fit the impedance plots in Figure 5d in the revised manuscript.

Reviewer 2 Report

Ms. No.:  nanomaterials-1691847

Title: Facile assembly of InVO4/TiO2 heterojunction for enhanced photo-oxidation of benzyl alcohol

Nanomaterials

The manuscript represents a study on the synthesis of hybrid catalysts based on InVO4/TiO2 for photodegradation of benzyl alcohol. The optical and semiconducting properties of synthesized materials were investigated. In general, the experimental results are presented more or less correctly and drawn conclusions are well documented. However, for the beneficial of the potential readers, some issues should be raised:

  1. The band gap energies from Tauc plots (as shown in Fig. 4b) were determined incorrectly due to a considerable absorbance at energies below Eg (high background). Please refer to the paper by Makuła, M. Pacia, W. Macyk, “How to correctly determine the band gap energy of modified semiconductor photocatalysts based on UV-Vis spectra” J. Phys. Chem. Lett. 9, 6814 – 6817 (2018).
  2. There in no information about the frequency at which Mott-Schottky plots (Fig. 4c and 4d) were constructed. Usually Mott-Schotky analysis should be performed for the whole spectrum of frequences or at least at a few one. There are missing conditions (electrolyte, temperature etc) at which these tests were performed. Finally, captions to Fig. 4c and 4d are completely not informative.
  3. Nyquist plots (Fig. 5c) should be drawn for the larger values of Z’. At least the whole spectra should be presented.
  4. There is no test for a simply photolysis of the used solution in Fig 6a. What a percent of benzyl alcohol degrade in the used conditions during illumination without any photocatalyst?
  5. Caption to fig 5 c and 5d is not informative. There is no information about a sample which was tested.
  6. A self-citation ration of manuscript authors is rather high. The reference list contains a lot of entries written by manuscript authors.
  7. Typos and errors in the manuscript should be corrected. There are sentences that start with a lowercase letter, missing spaces in the text, etc.
  1. The reference list should be carefully check and formatted/corrected. There is a lack of consistency in notation of article titles: upper or lowercase letters (e.g., compare ref. 11 and 12) are used. Ref. 4 is doubled as ref. 8. Ref. 21 the author name starts form the lowercase letter. Ref. 22 journal name written with lowercase letters. Ref. 29 the provided range pages is wrong.

To sum up, I think that the manuscript can be acceptable for publication after major revision.

Author Response

The band gap energies from Tauc plots (as shown in Fig. 4b) were determined incorrectly due to a considerable absorbance at energies below Eg (high background). Please refer to the paper by Makuła, M. Pacia, W. Macyk, “How to correctly determine the band gap energy of modified semiconductor photocatalysts based on UV-Vis spectra” J. Phys. Chem. Lett. 9, 6814 – 6817 (2018).

Response: Thanks for Reviewer’s suggestion. We have revised it according to the reference of J. Phys. Chem. Lett. 9, 6814 – 6817 (2018).

There in no information about the frequency at which Mott-Schottky plots (Fig. 4c and 4d) were constructed. Usually Mott-Schotky analysis should be performed for the whole spectrum of frequences or at least at a few one. There are missing conditions (electrolyte, temperature etc) at which these tests were performed.

Response: Thanks for Reviewer’s suggestion. We have added the test information in revised manuscript (The as-obtained samples were coated on FTO glass substrates to be used as the working electrodes. A 300 W Xe lamp with a cut-off filter (λ > 420 nm) was used as the light source and all of the electrochemical tests were carried out in KH2PO3-K2HPO3 buffer solution (0.5 M, pH=7).)

Finally, captions to Fig. 4c and 4d are completely not informative.

Response: Thanks for Reviewer’s suggestion. It has been added.

Nyquist plots (Fig. 5c) should be drawn for the larger values of Z’. At least the whole spectra should be presented.

Response: Thanks for Reviewer’s suggestion. It is done.

There is no test for a simply photolysis of the used solution in Fig 6a. What a percent of benzyl alcohol degrade in the used conditions during illumination without any photocatalyst?

Response: Thanks for Reviewer’s suggestion. we further investigated the effect of the used solution and light irradiation. It can be found that no BA conversion was found without light irradiation or catalysts).

Caption to fig 5 c and 5d is not informative. There is no information about a sample which was tested.

Response: It is done.

A self-citation ration of manuscript authors is rather high. The reference list contains a lot of entries written by manuscript authors.

Response: Thanks for Reviewer’s suggestion. We have changed the reference.

Typos and errors in the manuscript should be corrected. There are sentences that start with a lower case letter, missing spaces in the text, etc.

Response: It is done.

The reference list should be carefully check and formatted/corrected. There is a lack of consistency in notation of article titles: upper or lowercase letters (e.g., compare ref. 11 and 12) are used. Ref. 4 is doubled as ref. 8. Ref. 21 the author name starts form the lowercase letter. Ref. 22 journal name written with lowercase letters. Ref. 29 the provided range pages is wrong.

Response: It is done.

Reviewer 3 Report

This manuscript offers a concise report on assembly/construction of InVO4/TiO2 heterojunction for enhanced photo-oxidation of benzyl alcohol. The work elaborates on successful assembly of heterostructures/catalysts which, importantly in this case, is promoted by charge separation and transfer between InVO4 and TiO2. The authors also provide the adequate characterization effort by XRD, TEM and spectroscopy as well as the due analysis. From practical point of view, the reported results bring new knowledge about achieving a highly efficient photocatalyst via heterostructure assembly by relatively simple and easily available method which certainly is an original contribution in the present context.

Thus, the ambitious task in this work covers an array of hot lines of research of finding sustainable and simple ways to synthesize wide range photocatalysts in the shape of heterostructures with wide perspectives for groundbreaking applications that are currently attracting much research interest.

The authors chose an adequate structure of the manuscript – an excellent point of departure for such a study. Finally, the authors provided a balanced realistic and nicely illustrated presentation of their results and corresponding analysis that is of much scientific and practical interest and adds to new knowledge to the field.

In my opinion, the fine detailing in the present work, the insightful and balanced discussion of the results, as well as the very good figures, permit competent readers to utilize the manuscript as a guidance for future work. Consequently, this manuscript presents an efficient and beneficial basis for promoting and solving next step challenges in this field.

Moreover, the manuscript benefits from a clear motivation and it is an easy and informative read. The manuscript is also excellent in terms of clarity and accuracy of language.

The present manuscript is a significant contribution, this work once published would be quite useful as well as instructive and suggestive in terms of further studies and to a wider readership.

There are some minor issues with this already excellent manuscript that will need to be addressed before becoming suitable for publication, i.e., it can be considered for publication after a minor revision:

1: The authors miss part of bigger picture of assembly and synthesis of heterostructures and such cases whereby previous DFT studies of such systems have been widely used as accurate guidance to understand how they are assembled and controlled, e.g., The Journal of Physical Chemistry C 118 (2014), 5501-5509; and also Carbon 81 (2015) 620-628. Such works are supportive to the credibility of the searches and results as reported in the present manuscript.

2: The authors report characterization at room temperatures. Preparation of samples si discussed at temperatures of 120 deg. C There should be a clearer discussion what is the temperature range of the heterostructures considered here – synthesis temperatures, but also expected range of operational temperatures and also range of their thermal stability. Thermal stability would be addressable by molecular dynamics simulations by the way (I’m not suggesting doing that within present work).

3: The authors briefly comment Vanadium-Oxygen bonds in relation to spectroscopy. However, bonding at the interfaces needs to be commented/discussed in more detail.

4: Spell-check and stylistic revision of the paper are still necessary. Some long sentences, misspellings, etc., still are noticeable throughout the text.

Author Response

1: The authors miss part of bigger picture of assembly and synthesis of heterostructures and such cases whereby previous DFT studies of such systems have been widely used as accurate guidance to understand how they are assembled and controlled, e.g., The Journal of Physical Chemistry C 118 (2014), 5501-5509; and also Carbon 81 (2015) 620-628. Such works are supportive to the credibility of the searches and results as reported in the present manuscript.

Response: Thanks for Reviewer’s suggestion. It has been added.

2: The authors report characterization at room temperatures. Preparation of samples si discussed at temperatures of 120 deg. C There should be a clearer discussion what is the temperature range of the heterostructures considered here – synthesis temperatures, but also expected range of operational temperatures and also range of their thermal stability. Thermal stability would be addressable by molecular dynamics simulations by the way (I’m not suggesting doing that within present work).

Response: Thanks for Reviewer’s suggestion. The composite catalyst prepared at 120 ℃ shows the best performance. higher/lower temperature will destroy the heterojunction.

3: The authors briefly comment Vanadium-Oxygen bonds in relation to spectroscopy. However, bonding at the interfaces needs to be commented/discussed in more detail.

Response: Thanks for Reviewer’s suggestion. Vanadium-Oxygen bonds have been mentioned in the pure phase InVO4.

4: Spell-check and stylistic revision of the paper are still necessary. Some long sentences, misspellings, etc., still are noticeable throughout the text.

Response: Thanks for Reviewer’s suggestion. We have tried our best to polish the language in the revised manuscript.

Round 2

Reviewer 1 Report

The suggestions  and recommendations were considered by authors and the manuscript was modified accordingly.

Reviewer 2 Report

In the reviewed version of the manuscript the band gap energies from Tauc plots (as shown in Fig. 4b) were still determined incorrectly. Please,  take inco account a high absorbance at energies below Eg (high background). Please refer to the paper by Makuła, M. Pacia, W. Macyk, “How to correctly determine the band gap energy of modified semiconductor photocatalysts based on UV-Vis spectra” J. Phys. Chem. Lett. 9, 6814 – 6817 (2018).

Author Response

Response: Thanks for Reviewer’s suggestion. We have changed Tauc plots, and new  Tauc plots is shown in Figure 4b in the revised manuscripts.
